# PAX8 in the Junction between Development and Tumorigenesis

**DOI:** 10.3390/ijms23137410

**Published:** 2022-07-03

**Authors:** Reli Rachel Kakun, Zohar Melamed, Ruth Perets

**Affiliations:** 1Bruce and Ruth Rappaport Faculty of Medicine, Technion–Israel Institute of Technology, Haifa 3109601, Israel; r.kakun@gmail.com; 2Clinical Research Institute at Rambam, Rambam Health Care Campus, Haifa 3109601, Israel; 3Division of Oncology, Rambam Health Care Campus, Haifa 3109601, Israel; z_melamed@rambam.health.gov.il

**Keywords:** PAX8, embryogenesis, thyroid, kidney, uterus, fallopian tube, thyroid cancer, renal cancer, ovarian cancer, endometrial cancer

## Abstract

Normal processes of embryonic development and abnormal transformation to cancer have many parallels, and in fact many aberrant cancer cell capabilities are embryonic traits restored in a distorted, unorganized way. Some of these capabilities are cell autonomous, such as proliferation and resisting apoptosis, while others involve a complex interplay with other cells that drives significant changes in neighboring cells. The correlation between embryonic development and cancer is driven by shared proteins. Some embryonic proteins disappear after embryogenesis in adult differentiated cells and are restored in cancer, while others are retained in adult cells, acquiring new functions upon transformation to cancer. Many embryonic factors embraced by cancer cells are transcription factors; some are master regulators that play a major role in determining cell fate. The paired box (PAX) domain family of developmental transcription factors includes nine members involved in differentiation of various organs. All paired box domain proteins are involved in different cancer types carrying pro-tumorigenic or anti-tumorigenic roles. This review focuses on PAX8, a master regulator of transcription in embryonic development of the thyroid, kidney, and male and female genital tracts. We detail the role of PAX8 in each of these organ systems, describe its role during development and in the adult if known, and highlight its pro-tumorigenic role in cancers that emerge from PAX8 expressing organs.

## 1. Introduction

The understanding that high grade serous ovarian cancer (HGSOC), the most common subtype of ovarian cancer, originates from the fallopian tube epithelium, rather than from the ovary, has brought on a paradigm shift in ovarian cancer research, mainly in the search for prevention and early detection tools [1]. However, another important consequence of this discovery is the understanding that fallopian tube lineage markers might play a role in HGSOC. Soon afterward the fallopian tube emerged as the organ-of-origin of HGSOC, several manuscripts were published highlighting the critical role of PAX8 in HGSOC [2,3,4]. We were especially intrigued by the reactivation of a critical embryonic developmental factor in cancer. In this manuscript, we aim to explore the reactivation of this specific fallopian tube embryonic marker in the context of other developmental processes that are reactivated in cancer, and in the context of other PAX8 roles.

The many similarities between embryonic development and cancer progression are increasingly recognized, as these are both processes that require interaction and coordination among many cell types [5,6]. Cell proliferation, migration, and epithelial to mesenchymal transitions (EMT) are some of the shared processes [5]. Embryo implantation in the uterus is similar to cancer progression by utilizing processes such as invasion and angiogenesis [7]. Another common aspect is the immortality of cells; human preimplantation embryonic cells, similarly to cancer cells, are immortal and therefore can be indefinitely cultured in vitro [8,9]. An interesting concept when comparing development and cancer progression is that of organizing cells [6]. During development it is well established that in very early embryos there are “organizer” cells that are higher in the cell hierarchy and can influence other cells to change their differentiation [6,10]. In tumors, cancer cells command tissue microenvironment cells to assume tumor-supporting roles. The affected microenvironment cells include fibroblasts, nerve cells, adipocytes [11], immune cells [12], and endothelial cells [13]. Therefore, targeting of key cancer cell drivers affects not only the proliferation of cancer cells, but also curbs the supporting tumor microenvironment [12]. Examples of key intracellular drivers that also affect the microenvironment are KRAS [14], the ERK pathway, and intracellular transcription factors such as MYC [15], HIF-1 [16], and PAX8 [17,18]. Nevertheless, it is important to note that while these factors influence embryos to differentiate in an organized fashion, cancer progression is disorganized and includes dedifferentiation [6].

Many transcription factors, and transcription factor families, are involved in the regulation of embryonic differentiation and cancer transformation and progression. Some of these are factors that are expressed and play a key role in embryonic stages disappearing in adult cells and are reactivated in cancer. One example of such a gene is the well-known octamer-binding-protein, OCT4 (also known as OCT3), a product of the POU homeobox gene which is expressed in pre-implantation embryos and human tumors but not in normal somatic tissues [19]. OCT4 is specific to oocytes, the pluripotent cells of the embryo, and the germ line in humans and mice [20]. Similarly, OCT4 plays a major role in cancer stem cells, a subpopulation of cancer cells that is responsible for chemotherapy resistance [21,22].

Another group of transcription factors that are essential in embryonic development and tumorigenesis are the homeobox (HOX) genes [23,24]. The HOX genes play a central role in embryonic patterning and are responsible for the organization of the organs on the head–tail axis of the embryo. They are organized genomically in clusters residing in different chromosomes [25] and an altered expression of defined *HOX* clusters is found in different cancers, including alteration of *HOXA* genes in breast and cervical cancers, *HOXB* in colon cancer, *HOXC* in prostate and lung cancers, and *HOXD* in colon and breast cancer [26]. The involvement of *HOX* genes in cancer is therefore complex, with deregulation mechanisms of specific *HOX* genes differing among cancer types [27]. There is also a second class of homeobox genes that are not organized in these clusters and are called divergent genes [28]. The divergent genes also encode for transcription factors that play central roles in embryonic development and tumorigenesis [29]. Among them are the Distal-less homeobox family of genes [30], the NKX family of homeobox proteins, and the paired box domain family of proteins that will be detailed in the next section [31,32].

## 2. Paired Box Domain Family of Developmental Proteins

The paired box (PAX) domain proteins are homeobox divergent genes, which are vital regulators of embryogenesis, involved in tissue development and cell lineage specification and promote cell proliferation, migration, and survival [33]. Mutations and genetic aberrations in PAX family members are associated with extensive developmental defects and malignancies in humans and mice [33,34,35]. In mammals, there are nine PAX gene family members each encoding a nuclear transcription factor, and all are identified by a highly evolutionary conserved 128 amino acid paired box DNA-binding domain (Prd) at the N-terminal region of the protein [36].

The PAX family members are sub-classified into four paralog groups based on the presence or absence of a homeodomain (HD) region in the N-terminal DNA binding region and the presence of an octapeptide (OP) [37]. See Figure 1 for an illustration of PAX8 protein domains. Most PAX family members exist in several isoforms because of alternative splicing. The ability to produce different RNA transcripts, and thus several different protein structures from each gene, increases the functional diversity of PAX proteins. The expression of PAX proteins in adult tissues was recently summarized [38]. All PAX proteins play pro- or anti-tumorigenic roles in cancer. For a clear and succinct summary see [39].

## 3. PAX8 Is a Member of the Paired Box Domain Gene Family

The transcription factor PAX8 is a member of the paired box domain gene family, and, as with all paired box domain proteins, PAX8 expression is known to be essential in early embryonic development [42].

PAX8 carries an N-terminal DNA binding domain and a C-terminal transactivation domain. The DNA binding domain is composed of a paired box region, an octapeptide, and a truncated homeodomain. The paired box domain of PAX8 consists of two distinct subdomains known as PAI (N-terminal amino acids 1–72) and RED (C-terminal amino acids 77–128), each composed of three helices in a helix-turn-helix motif. Both subdomains are essential for DNA binding and a flexible linker connects these domains [43,44] (Figure 1). The gene encoding for PAX8 is located on chromosome 2q14.1 and has six splice variants, generated by alternative splicing (variants a–f). These isoforms exhibit different transcription properties but do not alter the DNA binding domain. PAX8A is the longest isoform, encoded by 12 exons, leading to a 450 amino acid protein. PAX8E and PAX8F were observed only in the placenta, where isoforms C and D are missing. Interestingly, the expression of isoforms E and F is abundant at the early embryonic stages and then gradually decreases. PAX8A becomes the predominant transcript in the late embryonic life of mice and in the post-embryonic life of mice and humans [45,46].

## 4. PAX8 in Thyroid Organogenesis and Cancer

### 4.1. The Role of PAX8 in Thyroid Organogenesis

The mammalian thyroid gland, located in the neck region, consists of two cell types: thyroid follicular cells (TFC) and C cells. The TFCs and C cells arise from different embryonic origins; TFCs originate from embryonic endodermal cells, and the source of C cells is neuro-ectodermal, originating from the neural crest at early embryogenesis [47]. TFCs are responsible for the formation of thyroid follicles that serve as storage and regulation structures of thyroid hormones (T_3_ and T_4_), while C cells are distributed in the inter-follicular space and produce calcitonin [48]. Mammalian thyroid differentiation is mainly governed by thyroid transcription factors (TTFs), namely NKX2.1, FOXE1, HHEX, and PAX8. These factors are detected in different embryonic tissues, although all four of them are synergistically expressed exclusively in thyroid follicular cells, suggesting that their co-expression is vital at early stages of thyroid morphogenesis [49].

The development of the murine thyroid begins on embryonic day 8 (E8), while PAX8 expression is first seen at E8.5 and maintained through the complete differentiation and organogenesis of the thyroid at E16.5. Pax8^−/−^ mice develop a normal primordial thyroid and budding (E9.5), but by E11.5 the primitive thyroid is smaller than its wild-type counterpart and a complete loss of thyroid follicular structures is seen by E12.5. Accordingly, Pax8^−/−^ mice show a severe hypothyroidism phenotype: thyroid hypoplasia, low birth weight, developmental delays, and overall high mortality rates during the first few weeks after birth [42]. The early presence of PAX8 preserves the survival of thyroid precursor cells by activation of the anti-apoptotic protein BCL2 [50]. However the development of the thyroid is not dependent on the expression of the anti-apoptotic protein BCL2 and likely also has other mechanisms since BCL2 knockout mice were shown to have a normal thyroid gland [51]. Another PAX8 transcriptional target that plays an important role in thyroid development is *Foxe1. Pax8*^−/−^ mice show a dramatic downregulation of FOXE1 expression on E9 as compared to WT mice, suggesting that *Foxe1* is a target gene of PAX8 in the early primordial thyroid [52]. However, while in *Foxe1* knockout mice the primordial thyroid cells do not migrate to the underlying mesenchyme (an essential process for thyroid formation), suggesting that FOXE1 is essential for cell migration, in *Pax8*^−/−^ mice the migration of thyroid primordial cells on E10 is similar to WT mice. This suggests that thyroid primordial cells can migrate to the underlying mesenchyme in a mechanism that is PAX8- and FOXE1-independent [52,53].

In humans, the development of the thyroid begins 20 days post fertilization (E20) when the thyroid anlage appears [54,55]. This is the first step in the process of specification that ultimately leads to the development of the thyroid. The thyroid anlage develops by thickening of the endodermal epithelium of the foregut, equivalent to mouse thyroid development on E8.5 [56]. In humans, on E24 the thyroid bud forms and starts to migrate towards the ultimate location of the thyroid near the trachea until the migration process completes on E45–E50 [57]. On E60, the forming thyroid fuses with the ultimobranchial bodies, which are the embryonic origin of the thyroid C-cells [57,58]. PAX8 expression is seen throughout the process of embryonic human thyroid formation, from specification to migration to the terminally differentiated thyroid phenotype by E70 [58]. PAX8 expression persists in adult thyroid follicular cells where it regulates the expression of many genes involved in the thyroid hormone-synthesizing machinery, such as thyroglobulin (TG), thyroid peroxidase (TPO), and sodium iodide symporter (NIS) [59,60]. Some patients with thyroid dysgenesis were found to have PAX8 heterozygous mutations. Most of the mutations were localized to the paired box domain, preventing binding to target sites and thus leading to a decreased expression of target genes [61,62,63]. Only few mutations were described in the promoter region of the PAX8 gene, explaining reduced protein expression [64,65]. In some of the PAX8 mutations DNA binding alternation is not the main defect, but rather the impaired synergism of the protein with other thyroid transcription factors, such as NKX2-1, or poor recruitment of transcriptional coactivators such as p300 [64,66,67].

The regulation of PAX8 in thyroid development is unclear. In adult cells, a complex network for the regulation of PAX8 expression and activity is seen. For example, the Hippo pathway mediator TAZ was shown to positively and negatively regulate the transcriptional activity of PAX8 on different promoters [68,69]. Interestingly, TAZ also regulates the activity of another thyroid embryonic factor, TTF1 [69].

### 4.2. The Role of PAX8 in Thyroid Carcinoma

There are several subtypes of epithelial thyroid cancers that arise from the follicular cells of the thyroid. These are divided into differentiated thyroid cancers, which are the majority, and undifferentiated cancers, that are much more aggressive [70]. Differentiated thyroid cancers are further classified into papillary thyroid cancers (PTC), which are about 70–80% of differentiated thyroid cancers, follicular thyroid carcinomas (FTC), comprising about 10–15% of differentiated thyroid tumors, and a small minority of other rare subtypes [71,72,73]. Undifferentiated thyroid cancers are mostly comprised of anaplastic thyroid cancers, which are less than 2% of the thyroid cancers that arise from the follicular cells of the thyroid [74].

PAX8 expression can be found in 100% of papillary thyroid carcinomas, in 91–100% of thyroid follicular carcinomas, and in variable levels of up to 80% in anaplastic thyroid carcinomas [75,76,77]. Furthermore, 30% of follicular thyroid carcinomas contain a balanced translocation, t(2;3) (q13;p25), that results in fusion of the PAX8 and peroxisome proliferator-activated receptor γ (*PPARG*) genes with concomitant expression of a PAX8–PPARγ fusion protein (PPFP). PPFP can also be found in the follicular variant of papillary thyroid carcinoma, albeit at a much lower rate [78].

PPARγ encodes for a nuclear hormone receptor transcription factor whose activity is related to adipocyte differentiation, lipid and carbohydrate metabolism, and cellular proliferation and differentiation [71,79]. PPARγ is expressed at very low levels in normal thyroids and has no known function in that organ. PPFP demonstrates a well-documented oncogenic capacity through a mechanism in which it acts as a dominant-negative inhibitor of wild-type PPARγ. Likely the PAX8 part of PPFP does not affect the function of PPFP, but rather the high expression of this fusion protein is a result of the PAX8 promoter [80].

The role of PAX8 itself in thyroid carcinomas that do not harbor the PPFP fusion protein is unclear. PAX8 target genes were shown to be dysregulated in PTC, but the biological implications of these changes were not tested [81]. BCL2 was shown to be a PAX8 target gene during embryonic development, but there is no evidence suggesting that PAX8 regulates BCL2 in thyroid cancer, although both genes are co-regulated by the same microRNA [82]. FOXE1 is another suggested PAX8 target gene in thyroid development that to the best of our knowledge was not shown to be a PAX8 target gene in thyroid cancers. The overall differences and similarities between PAX8 target genes in thyroid development, adult thyroids, and thyroid cancers remain to be shown.

TTF1 and PAX8 are both markers of embryonic thyroid development that are often co-expressed in thyroid cancer cells, and TTF1 was shown to regulate the expression of PAX8 in thyroid cancer cells. Of interest, the effect of PAX8 expression of cancer cell proliferation and migration is relatively minor, and TTF1 has a more pronounced effect on the tumorigenic properties of thyroid cancer cells [83].

## 5. PAX8 in Kidney Development and Cancer

### 5.1. The Role of PAX8 in Kidney Development

The renal system is composed of the kidney and the urogenital tract. Renal organogenesis is a step-wise process that starts from the development of the pronephros arising from the intermediate mesoderm on E8.5 of mice, turning into the mesonephros and then the metanephros, the latter of which initiates development on E11.5 in mice [84,85] (Figure 2). Kidney organogenesis begins on E8.5 when the intermediate mesoderm undergoes mesenchymal-to-epithelial transition (MET) that gives rise to the pronephros, also termed nephric (Wolffian) duct [84]. The nephric duct then elongates caudally during the transformation of the pronephros to the mesonephros, and as it elongates it induces the formation of mesonephric tubules from the adjacent intermediate mesoderm (also called nephrogenic cord or nephrogenic mesenchyme) [86]. Caudally, the nephric ducts fuse with urogenital sinus (the ventral part of the cloaca), the latter giving rise to the adult bladder [87]. Around E10.5, the caudal part of the pronephros (Wolffian duct) gives rise to a branching outgrowth called the ureteric bud [86].

The ureteric bud grows into a special part of the nephric cord called the metanephric mesenchyme, and the ureteric bud induces the metanephric mesenchyme to turn into metanephric nephrons that will eventually give rise to the kidney [88]. The ureteric bud itself develops into the urogenital tract. The mesonephros and the mesonephric tubules start degenerating on E14.5 and almost entirely disappear in 24 h [89,90]. In females, the mesonephros entirely disappears, while in males some of the cranial mesonephric tubules turn into the epididymal ducts of the testes [85].

PAX8 expression is detected in renal development even before the pronephros is formed, around E7–8 in mice (6–7 somite stage), and is retained during embryogenesis all the way through the developmental process until the adult kidney is formed [91]. However, Pax8 knockout mice show no kidney abnormalities, suggesting that PAX8 expression is redundant in the kidney [42]. Another PAX protein that plays a role in kidney development is the PAX8 close family member PAX2. PAX2 is expressed in the intermediate mesoderm during the nephric anlage stage (before pronephros formation) in the 8–9 somite stage, slightly later then PAX8 expression [91]. However, in PAX2 knockout mice, the pronephros and mesonephros are formed but the metanephros fails to form, resulting in a lack of kidney and genital tract formation [92,93,94]. Combined PAX2 and PAX8 knockout mice show a more severe kidney phenotype, where even the pronephros and mesonephros are not formed. This suggests that PAX8 can compensate for the lack of PAX2 during pronephros and mesonephros development [91]. Of note, while the upper collecting system is PAX8-positive and stems from the metanephros, the bladder stems from differentiation of the urogenital system and is accordingly PAX8-negative [95].

There is very little information regarding the role of PAX8 in the adult renal kidney, and likely most of its role is shared with PAX2. However, it was shown that in inner medullary collecting duct epithelia of adult mice that PAX8, but not PAX2, binds the promoter of *Slc14a2*. *Slc14a2* is the gene encoding for urea transporters and regulating the expression of urea transporters in response to high salt conditions to maintain salt homeostasis [96].

**Figure 2 ijms-23-07410-f002:**
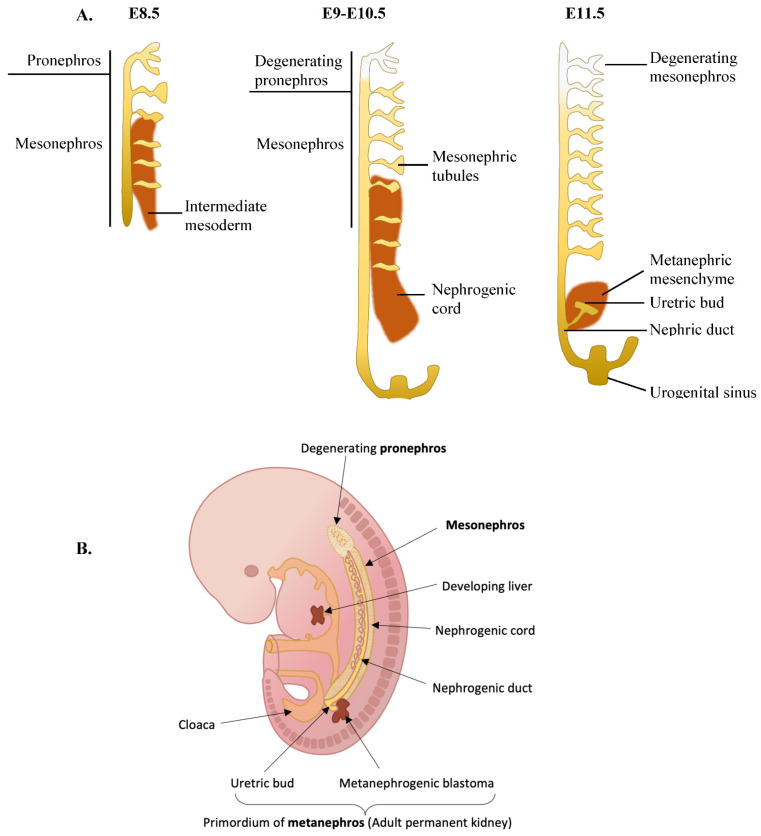
Illustration of embryonic kidney development in mice. (**A**) An illustration of the kidney development timeline in mice. (**B**) Mouse kidney development on E11 shown in the context of the whole mouse embryo [97].

### 5.2. The Role of PAX8 in Renal Cancers

PAX8 is expressed in several tumors of the urogenital system, such as in the majority of pediatric Wilms tumors [98,99,100], ~80% of renal cell carcinomas (RCC), and nephrogenic adenomas. Tumors of the collecting system diverge in their PAX8 expression. The upper part of the collecting system stems from the PAX8-positive metanephros, and accordingly 23% of tumors of the upper part, specifically of the renal pelvis, are PAX8-positive. The lower part of the collecting system arises from the PAX8-negative urogenital sinus, and tumors that arise from the lower part, i.e., tumors of the ureters and bladder, are all PAX8-negative [77,95,101].

There is a paucity of information regarding the role of PAX8 in RCC. PAX8 plays a positive proliferative role in RCC, and PAX8 knockdown leads to growth arrest and senescence [102,103]. This induction of growth arrest occurs via two different mechanisms. First, PAX8 knockdown was shown to transcriptionally reduce E2F1 expression and reduce the expression of several E2F1 target genes leading to E2F1-dependent growth arrest. Interestingly, and perhaps paradoxically, PAX8 was also shown to stabilize protein levels of RB, the negative regulator of E2F1 [103]. This suggests that PAX8 can perhaps regulate E2F1 levels both positively in a direct manner and negatively in an indirect manner. Another mechanism that was recently suggested for the positive role of PAX8 on cell proliferation is activation of various metabolic pathways in RCC, among them ferroxidase ceruloplasmin, thus altering the cell’s iron metabolism [102].

## 6. PAX8 Expression in Female Reproductive Tract Development and Cancer

### 6.1. The Role of PAX8 in Female Reproductive Tract Organogenesis

In early embryonic stages, the urogenital system is derived from the intermediate mesoderm of the gastrula [104]. The Müllerian ducts develop in close proximity to the Wolffian ducts, which eventually form the urinary tract [105]. In humans, the two Müllerian ducts begin to migrate medially and then fuse and descend into the pelvis at around 6 weeks of gestational age [106]. Fusion is generally complete by 14 weeks of gestational age, at which time the structures canalize [106,107]. During this time, the urogenital sinus invaginates and differentiates into the sinovaginal bulb [108]. At approximately 12 weeks of gestation, the sinovaginal bulb grows cephalad and fuses with the most caudal portion of the Müllerian ducts [109,110] (Figure 3). In mice, a similar process occurs where the Müllerian duct starts to form by invagination of the surface epithelium of the anterior mesonephros (embryonic kidney) around E11.5 [111]. The Müllerian duct then elongates until it is completed by E13.5, when the duct has reached and conjoins the urogenital sinus [111,112].

In the murine Müllerian duct, PAX8 is detected as early as E11.5–E12.5 [113,114]. Although it has been shown to be expressed in the entire Müllerian duct, PAX8 knockout mice lack endometrial development; however, there is normal development of other structures that arise from the Müllerian duct, such as the upper part of the vagina, cervix, and oviduct [115,116]. The latter suggests that other transcription factors compensate for the absence of PAX8 in these structures [115]. In the adult human uterus, PAX8 is expressed in all epithelial cells of the endometrium. In the fallopian tube, PAX8 is expressed in the secretory epithelial cells, but not in ciliary epithelial cells [77]. In the cervix, PAX8 is expressed in the glandular cells of the uterine cervix (endocervix) whilst not detected in the squamous epithelial cells (ectocervix) [77]. The role of PAX8 in the normal adult female genital tract is unknown.

### 6.2. The Role of PAX8 in Tumors of the Female Genital System

Tumors arising from the Müllerian system are divided into uterine and extra uterine cancers [117]. The most common female genital tract extra-uterine cancer is epithelial ovarian cancer (EOC). The most common subtype of EOC is high-grade serous carcinoma (HGSC), which in the majority of cases arises from the fallopian tube epithelium [1].

PAX8 was suggested in recent years to be a highly specific marker for identification of Müllerian tumors, both uterine and extra-uterine [116,118]. Different studies have shown that PAX8 is expressed in 89–92% of EOC [77,119,120] and 98–100% of endometrial tumors [77,118]. Of note, all non-mucinous EOC subtypes are PAX8-positive in the vast majority of tumors, and only mucinous ovarian carcinomas are mostly PAX8-negative, suggesting a different embryonic origin [77,121]. In uterine mixed Müllerian tumors (MMMT), which encompass both epithelial and sarcomatous components, PAX8 is expressed in the epithelial component of 97% of all uterine MMMTs but is much less common in the sarcomatous component [122]. For cervical cancer, there is consensus that squamous cell carcinomas of the cervix do not express PAX8, but there is a range in the rate of adenocarcinomas of the cervix that are positive, from 83% to none, likely due to small sample sizes [77,123]. Therefore, PAX8 is clinically used as a good IHC marker for diagnosing Müllerian tumors. One study calculated the sensitivity of PAX8 staining for diagnoses as ranging from moderate to high sensitivity, with high specificity for diagnosing carcinomas of Müllerian origin [124].

PAX8 plays many oncogenic roles in different subtypes of EOC. We and others have shown that PAX8 inhibits HGSC cell proliferation via inhibition of apoptosis [2,3,125]. We have shown that PAX8 activates mutated oncogenic p53 and pro-proliferative cytoplasmic p21, together driving the pro-proliferative effect of PAX8 [3]. Regulation of apoptosis by PAX8 was also shown in a FOXM1-dependent mechanism [125]. A recent study suggests that PAX8 has a pro-angiogenic role in HGSCs via its interaction with another developmental factor, SOX17. PAX8 and SOX17 are co-expressed in ovarian cancer cells [126] and together inhibit the expression of *SERPINE1,* an anti-angiogenic factor, thereby promoting angiogenesis [18]. The PAX8 and SOX17 complex was also shown to regulate other genes involved in the cell cycle and morphogenesis. Interestingly, PAX8 and SOX17 were shown to be co-regulated by CDK12/13, and biochemical inhibition of CDK12/13 downregulates the expression of both proteins [126]. Another work has shown that PAX8 cooperates with PRDM3 (the protein product of the MECOM genomic locus) to regulate genes that are involved in adhesion and the extracellular matrix [127]. The role of PAX8 in HGSC migration was also shown to be mediated by PKCα [17]. Another role for PAX8 is binding of YAP1/TEAD proteins and transcriptional activation of the Hippo/YAP signaling pathway [128], known to regulate pro-survival genes [129]. Taken together, the information regarding the role of PAX8 in HGSCs fits its definition as a master regulator transcription factor of HGSCs, regulating about 30,000 genes and governing many hallmarks of cancer such as evading apoptosis, angiogenesis, and migration [2,3,17,18,127].

In non-HGSC EOC subtypes, PAX8 was shown to promote anchorage-independent growth, likely by increasing cellular adhesion, thereby contributing to EOC tumorigenicity [130]. Furthermore, PAX8 was shown to have a role in the migration and invasion of EOC [4]. Other studies have shown that in non-HGSC EOC, PAX8 exerts its pro-proliferative role via inhibition of growth arrest and subsequent transcriptional induction of E2F1 expression [103]. Overall, the role of PAX8 in HGSCs and non-HGSC EOC is similar, and different from the minor role of PAX8 in benign cells of fallopian tube lineage [128].

Similar to the non-essential role of PAX8 in fallopian tube development [115], it seems to be also non-essential in adult fallopian tube secretory epithelial cells (FTSEC). However, when FTSEC transform, the role of PAX8 becomes essential to these cells and the number of target genes significantly rises. It is unclear what the exact mechanism is that leads to the major change in the role of PAX8 during transformation. Some suggested mechanisms are epigenomic changes [128], or changes in the role of its target genes, such as a mutation in p53 and an aberrant role of p21 [3].

Another interesting question is the source of PAX8 expression in non-HGSC EOC. PAX8 is a lineage marker of FTSEC, and therefore it is not surprising that it can be found in HGSCs, which arise from transformation of FTSEC. However, non-HGSC EOC is not considered to arise from FTSEC, which raises a question regarding the origin of PAX8 expression in these cells. Interestingly, endometrioid and clear cell carcinomas also express PAX8 in the majority of tumors [77], although these subtypes do not originate from PAX8-positive FTSEC. It was proposed that these tumor types arise from endometriosis, PAX8-positive endometrial cells ectopically expressed outside the uterus [131,132]. Alternatively, these tumors may arise from ovarian surface epithelial cells, and there have been reports of PAX8 expression in ovarian surface epithelium [133]. However, ovarian surface epithelial cells are more commonly considered PAX8-negative, and occasional PAX8-positive cells in the ovarian surface epithelium are thought to be FTSEC or endometrial cells that migrate to the ovarian surface [116,134], leaving PAX8-positive endometriosis cells as the main cell of origin for clear cell and endometrioid ovarian cancers.

In endometrial cancer, PAX8 is expressed in over 90% of tumors of endometrioid and serous histologies [77]. We have published on the role of PAX8 in an especially aggressive subtype of endometrial cancer, uterine serous papillary carcinoma. We have shown that in uterine serous papillary carcinoma, PAX8 plays a pro-proliferative and anti-apoptotic role that is mediated via transcriptional activation of cytoplasmic p21, which has an anti-apoptotic role in this disease [135].

## 7. PAX8 in the Male Reproductive System—Development and Cancer

### 7.1. The Role of PAX8 in Male Reproductive System Organogenesis

The main purpose of the male reproductive system is generation, storage, and ejaculation of sperm as well as the production of androgens. Spermatogenesis occurs in the seminiferous tubules of the testes. Sperm maturation mostly occurs in the periphery of the seminiferous tubules. As the sperm matures it moves into the lumen of the tubules and then, via a net of small tubes called the rete testes, through small ducts, called efferent ductules, and into the epididymis. The sperm matures and becomes motile in the epididymis and eventually, during sexual arousal, moves into the vas-deferens where it is combined with fluid coming from the seminal vesicles in the ejaculatory duct, and from there both sperm and fluid are ejaculated via the urethra [136]. The testes, epididymis, vas deferens, ejaculatory ducts, and seminal vesicles are all derived from the Wolffian (mesonephric) duct, which degenerates in the female embryo (Figure 3) [137]. The testes also contain Leydig cells which produce testosterone, the major driver of male genital tract differentiation [136].

In mice, around E13.5, the mesonephric tubules develop into the efferent ductules that connect the rete testis with the upper part of the epididymis [138]. PAX8 expression is detectable at E9.5 in the developing mesonephric duct and also later at E15.5 in the epididymis of male mice [139]. PAX8 expression is also seen in progenitor cells of the rete testis as early as E10.5, showing that the rete testes are also partly derived from the mesonephric duct [140].

Male *Pax8* knockout mice are infertile due to malformation and obstruction of the genital tracts. Specifically, *Pax8* knockout mice have either agenesis or significant degeneration of the efferent ductules and epididymis. The lack of drainage of the testes results in secondary disorganization of the seminiferous epithelium and atrophy of the testes that progresses with age, likely due to sustained pressure from the drainage blockade. This suggests a role for PAX8 in embryonic development of the efferent ducts and epididymis. In *Pax8* knockout mice, there was also some degeneration in the development of the vas deferens, another mesonephros derived organ. However, since overall organs that derive from Wolffian structures develop, it is likely that, in early stages of male genital tract development, the loss of PAX8 is compensated by other factors, such as PAX2. In later stages however, when proper morphogenesis of the efferent ductules and the epididymis occurs, the role of PAX8 is critical. This is also evident from the finding that the seminal vesicles, which are also PAX8-positive and derived from the mesonephric duct, are intact in *PAX8* knockout mice. Of note, the development of the seminal vesicles is androgen-dependent and is intact, suggesting proper hormonal function in these mice [141].

In adulthood, in mice and humans, PAX8 is expressed in the epithelium of the seminal vesicles, epididymis, testicular efferent ductules, and rete testes, but not in the urothelium, germ cells, Leydig cells, Sertoli cells, or the prostatic epithelium [142]. These distinctive patterns are probably related to the different embryonic origins of these organs that do not originate in the mesonephric ducts [143,144].

### 7.2. The Role of PAX8 in Male Genital Tract Neoplasms

A limited number of male genital tract epithelial tumors show PAX8 immuno-positivity, all of which are rare tumors. PAX8-positive staining can be found in primary epithelial tumors that result from transformation of Wolffian derived organs. Specifically, PAX8 expression is seen in serous cystadenoma of the epididymis, carcinoma of the rete testis, Wolffian adnexal tumors, and endometrioid carcinoma of the seminal vesicle [142,145].

PAX8 staining can aid in differentiating these tumors from other male genitourinary tumors that are PAX8-negative, such as urinary tract, testicular, and prostate cancer [142].

## 8. PAX8 in Other Organs

Over the years, several studies looked for PAX8 expression in other organs that was not the well-characterized expression found in the thyroid, kidney and genital tracts. However, most of these studies used a polyclonal antibody for PAX8 (Proteintech, cat #10336) which was later found to be cross-reactive with other PAX family members such as PAX2, PAX5, and PAX6 [146,147]. The high homology of the N-terminal paired domain of all PAX family members renders N-terminal antibodies, or polyclonal antibodies that detect a large fragment of the protein, to be specifically prone to such cross-reactivity [148]. Therefore, in studies looking for PAX8 expression in organs previously unknown to express the protein, either a C-terminal antibody should be used [149] or mRNA expression should be tested. Western blot analysis using siRNA is another useful tool to determine antibody specificity.

In light of these caveats, the literature pertaining to PAX8 expression in other organs should be critically addressed. For example, the involvement of PAX8 in normal pancreatic islet cells was suggested using the aforementioned PAX8 polyclonal antibody [142]. However, other publications have shown that a more specific PAX8 antibody does not detect PAX8 in developing or adult islet cells, that PAX8 mRNA is not expressed in adult islet cells, and that the antibody that was used to detect PAX8 in islet cells is cross-reactive with PAX6 [150]. Of note, although PAX8 is not expressed in normal pancreatic islet cells when tested using appropriate methods, its expression is activated during pregnancy [151]. Interestingly, a father and daughter carrying a germline PAX8 mutation were described. The father had suffered from type 2 diabetes, while the daughter developed gestational diabetes mellitus, further implicating PAX8 in pancreatic islet cell function during pregnancy [152]. Another paper described a single nucleotide polymorphism (SNP) in the PAX8 locus in Afro-Americans that correlated with elevated risk of type II diabetes [153]. Therefore, while PAX8 is not reliably detected in normal islet cells, it can still be implicated in islet function under certain conditions.

Accordingly, the expression of PAX8 was described in pancreatic neuroendocrine tumors (PNET) [142,154,155] but disputed by others based on antibody cross-reactivity [149,150,156]. Staining of PAX8 in PNET was clearly shown to be a result of antibody cross-reactivity with another PAX family member, PAX6 [150,156].

PAX8 is expressed in the developing brain, specifically in the forebrain and hindbrain, but disappears in the adult brain [157,158,159]. In terms of the function of PAX8 in the developing brain, PAX8 knockout mice did not show brain abnormalities [42,160]. PAX8 does plays a role in mouse inner ear morphogenesis and development, but this role is non-essential since PAX8 knockout mice have no defects in inner ear development [161].

In brain cancers, PAX8 was shown to be expressed in medulloblastoma but only in the subtypes associated with sonic hedgehog and wingless signaling, and not in groups 3 or 4 [159]. Interestingly, PAX8 expression in medulloblastoma was an independent marker of good prognosis, and PAX8 knockdown reduced proliferation, a reverse role compared to its known role in Müllerian tumors. The expression of PAX8 was also studied in glioblastoma, but with contradictory results. Khanlou et al. showed that in glioblastoma sections PAX8 expression was seen focally in only one of nine examined cores [158], while Hung et al. showed PAX8 expression in the vast majority of glioblastoma samples. Of importance, Hung et al. detected the expression of PAX8 protein using two different antibodies, including a C-terminal protein that likely is not cross-reactive with other PAX proteins, and using reverse transcription polymerase chain reaction (RT-PCR), therefore excluding antibody cross-reactivity as the basis for the results. PAX8 expression was also seen rarely in low-grade astrocytomas and grade 1 meningioma tumors. Malignant meningioma (grade 3) exhibited PAX8 expression in 4/4 examined tumors [162].

Two comprehensive immunohistochemical studies reported expression of PAX8 in 100% of a sub population of lymphocytes (presumably B cells) in normal lymphoid tissues. In addition, positive PAX8 diffuse staining has been observed in 100% of the tested lymphomas. However, this study used a polyclonal PAX8 antibody (Proteintech cat #10336) which was subsequently shown to be cross-reactive with PAX5 that is known to be expressed in B-cells. Accordingly, PAX8 was not detected in normal B-cells nor in B-cell lymphomas using a PAX8 C-terminal antibody that does not cross-react with PAX5. Furthermore, PAX8 mRNA levels were not detected in any of the B-cell lymphoma cell lines studied [147].

This example highlights the importance of using specific antibodies when reporting PAX8 expression in different tissues.

## 9. Summary

PAX8 is a lineage marker with important developmental roles in the thyroid, Wolffian, and Müllerian systems in mice and humans. The expression of PAX8 persists in adult tissues but the role of PAX8 in adult non-thyroid tissues is unclear. When PAX8 expressing cells are transformed, PAX8 gains a new oncogenic role. There is debate regarding the expression of PAX8 in other tissues such as the brain, where expression subsides in the adult brain and turns back upon transformation. The comparison of PAX8 expression and its role in development and cancerous tissues highlights the connections between embryonic development and cancer formation.

## Figures and Tables

**Figure 1 ijms-23-07410-f001:**
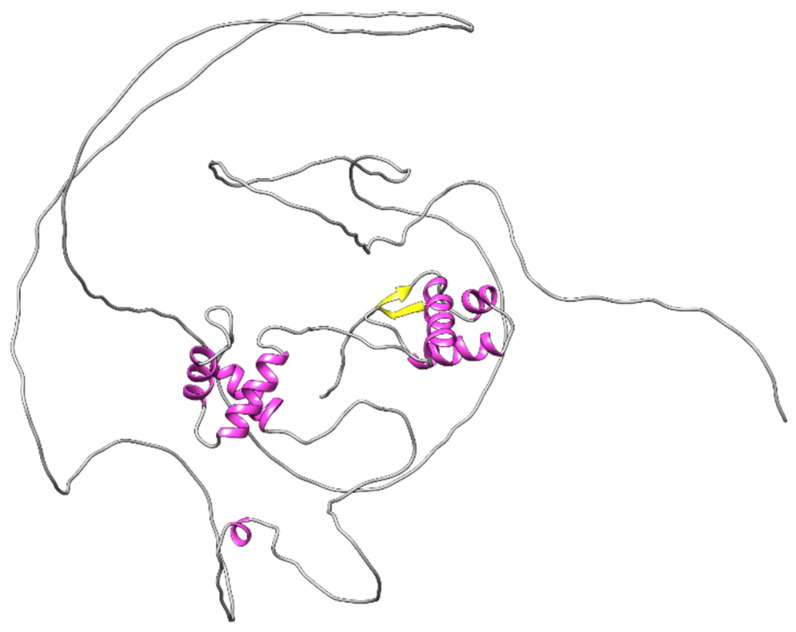
Alpha fold prediction of the structural organization of the PAX8 protein. The PAI and RED subdomains that are composed of helix-turn-helix motifs are seen in pink. A β hairpin is seen in yellow at the N-terminus of the PAI subdomain. Both PAI and RED subdomains are predicted to bind DNA. The octapeptide, homeodomain, and transactivation domains are predicted to be disorganized structures [40,41].

**Figure 3 ijms-23-07410-f003:**
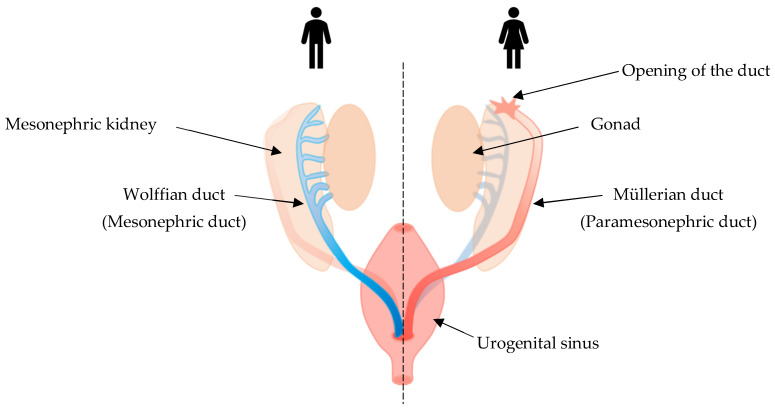
Illustration of the development of the male and female genital tract. Degenerating ducts are shaded.

## Data Availability

Not applicable.

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
