# Peer review of "PAX8 in the Junction between Development and Tumorigenesis"

_ijms, 2022, doi:10.3390/ijms23137410_

Round 1
Reviewer 1 Report
The submitted manuscript entitled “PAX8 in the junction between development and tumorigenesis” focuses on the discussion of the roles of PAX8 in embryonic development and cancer progression. The manuscript is well-organized and informative. I have some recommendations to improve the quality of the manuscript.
1. The usage of terms “embryonic” and “embryonal” should be unified. The “embryonic” is better.
2. In the Introduction, a rationale to discuss a role of PAX8 should be provided. Why PAX8?
3. In section 2, lines 74-89, a Figure to depict structural organization of PAX proteins is required.
4. In section 3, lines 90-107, a Figure to illustrate the mechanism of gene expression regulation by PAX transcription factors is recommended.
5. Since in many studies the co-expression of PAX8 and other transcription factors such as COX17 and TTF-1 in cancer have been reported, it would be useful to discuss this issue in more details. See, for example, https://doi.org/10.1038/s41388-022-02210-3 and https://doi.org/10.3892/ijo.2016.3617, etc.
6. English language style should be checked and edited.
Reviewer 2 Report
This review reviewed the Paired box (PAX) family of developmental transcription factors, which includes nine members that are involved in the differentiation of various organs. Accordingly, all members of the PAX family are involved in various types of cancer, some of which have pro-oncogenic and some antitumor functions. This review focuses on one specific member of this family, PAX8, which is a major transcriptional regulator in the embryonic development of the thyroid, kidney, and male and female reproductive tracts. In this review article, the authors detail the role of PAX8 in each of these organ systems and highlight its pro-oncogenic role in cancers arising from PAX8-expressing organs. The review is detailed, well structured, contains detailed information and well-chosen illustrative material. I think that the article deserves to be accepted for publication in its present form.
Author Response
Thank you for your favourable opinion of our manuscript.
Round 2
Reviewer 1 Report
The authors have addressed all my critical concerns.